# Alcohol, Tobacco and Cannabis Consumption on Physical Activity and Physical and Social Self-Concept in Secondary School Students: An Explanatory Model Regarding Gender

**DOI:** 10.3390/ijerph191610243

**Published:** 2022-08-18

**Authors:** Eduardo Melguizo-Ibáñez, Félix Zurita-Ortega, Gabriel González-Valero, Pilar Puertas-Molero, Georgian Badicu, Gianpiero Greco, Stefania Cataldi, Francesco Fischetti

**Affiliations:** 1Faculty of Education Sciences, Department of Didactics of Musical, Plastic and Corporal Expression, University of Granada, 18071 Granada, Spain; 2Department of Physical Education and Special Motricity, Faculty of Physical Education and Mountain Sports, Transilvania University of Brasov, 500068 Brasov, Romania; 3Department of Basic Medical Sciences, Neuroscience and Sense Organs, University of Study of Bari, 70124 Bari, Italy

**Keywords:** harmful substances, psychosocial aspects, active lifestyle, secondary school education, teenagers

## Abstract

Nowadays, the adolescent population consumes substances that are harmful to health at an earlier age. Therefore, the present research aimed to (i) develop an explanatory model of tobacco, alcohol, and cannabis consumption on physical self-concept, social self-concept, and physical activity practice and (ii) contrast the model through a multi-group analysis according to the gender of the participants. For this purpose, descriptive, comparative, and cross-sectional research was carried out on adolescent students (M = 13.91; SD = 1.31, years), using the Self-Concept Form 5 Questionnaire, the Physical Activity Questionnaire for Adolescents (PAQ-A), and the State Survey on Drug Use in Secondary Education (ETUDES) for data collection. Findings revealed that the consumption of harmful substances has a positive impact on the social area and the practice of physical exercise, showing a negative relationship between the latter variable and the social and physical area of self-concept.

## 1. Introduction

Adolescence is a period of development and growth between childhood and adulthood [1]. This stage of development is one of the most crucial in human development, not only because of the social and psychological changes that occur but also because of the acquisition of numerous patterns that tend to persist into adulthood [2]. Similarly, gender plays a major role during adolescence, as there are significant differences between the development of boys and girls [2,3].

During adolescence, the social influences of the peer group or friends, together with other variables such as family and school, can have an impact on the consumption of harmful substances [4]. During adolescence, the onset of alcohol consumption is more rapid, mainly due to peer group or social group membership [5]. Alcohol consumption can lead to numerous health problems such as problems related to bone density, reproduction, and even fertility [6,7]. Tobacco use may also begin during this stage, as this substance acts as a facilitator of social interaction and is socially approved [8,9]; however, tobacco use has become a public health problem, as it is occurring earlier in young adolescents. The onset of alcohol and tobacco consumption may precede the consumption of substances such as cannabis, with numerous studies [10,11,12] affirming the effects of this substance on the brain, from a functional and structural point of view. Moreover, not only harmful effects are reflected not only in the brain but also in the respiratory, cardiovascular, social, and psychotic areas [13].

Likewise, during adolescence, the construction and development of personality play a fundamental role in belonging to a particular social or peer group [14]. Exclusion for not belonging to a certain social group can have a negative impact on social self-concept, which can be defined as the perception that each subject has of themselves regarding their social skills when interacting with their peers [15]. The main reasons for exclusion from a particular social group during adolescence stem from physical appearance [16]. It has been shown that teenagers who are overweight or obese tend to be excluded because of their fitness [17]. Given these results, it has been found in the scientific literature that adolescents with higher physical self-concept scores show better integration into different social groups, with those with lower levels in this area of self-concept presenting greater difficulties [18], with these low scores being mainly due to lower physical activity times [19].

In adolescence, it has been shown that there is a decrease in the amount of time spent doing physical activity as young people move towards more sedentary activities [20]. Regular physical exercise has been shown to have numerous physical and mental health benefits, including improvements in blood pressure, as well as numerous bone, cognitive, and social benefits [21,22]. Improvements in physical self-image have also been found [23,24], exerting a positive effect on adolescents’ self-concepts.

Taking into account the above, a study is presented here that relates the consumption of harmful substances with psychosocial aspects (physical self-concept and social self-concept) and the effect of these variables on the regular practice of physical activity. In order to detail the research in a more precise way, it is structured in the following sections: Materials and Methods, where the characteristics of the participants, the design of the research, the instruments used, and the data analysis are carried out are contextualized. Next comes Section 3, in which the analyses carried out to respond to the research objectives and hypotheses can be found. This is followed by the discussion of the results, which shows the objective of comparing the results obtained with those of other research studies. This section is followed by the section on limitations and future perspectives, where the limitations of the research are pointed out and, finally, the conclusions of the study are observed. 

Although the research topic of psychosocial aspects and substance use has been studied and addressed in the scientific literature [4,6,9], the present research develops an explanatory model according to the gender of the participants. Through this model, it is possible to study the existing relationships between the variables that make up each model, providing greater precision in the study of the effect of some variables on others. Therefore, this research reflects the following research hypotheses:

**H.1.** 
*Differences are expected to be found between the male and female gender.*


**H.2.** 
*The consumption of alcohol, tobacco, and cannabis will have a negative impact on physical self-concept and the practice of physical activity.*


**H.3.** 
*Harmful substance use will have a positive impact on social self-concept.*


**H.4.** 
*Physical self-concept will be positively related to physical activity.*


**H.5.** 
*Physical activity will be positively associated with social self-concept.*


Finally, the research objectives presented are as follows: To study the levels of consumption of alcohol, tobacco, cannabis, social and physical self-concept, and physical activity practice; to develop an explanatory model of tobacco, alcohol, and cannabis consumption on physical self-concept, social self-concept, and physical activity practice; and to contrast the model through a multi-group analysis according to the gender of the participants.

## 2. Materials and Methods

### 2.1. Design and Participants

A descriptive, comparative, cross-sectional, non-experimental (ex post facto) study was carried out with some secondary school students in the province of Granada, with these schools being randomly selected. The total sample consisted of 706 participants (M = 13.91; SD = 1.31), of which 56.1% (*n* = 396) were male and 43.9% (*n* = 310) were female. In terms of sampling error, a sampling error of less than 5.0% was set for a confidence level of 99%.

### 2.2. Instruments

Sociodemographic Questionnaire: Aimed at collecting sociodemographic variables such as gender and age of the participants.

Self-Concept Questionnaire Form-5 [25]: Such an instrument was designed to measure self-concept from a penta-dimensional perspective, namely, academic self-concept (items 1, 6, 11, 16, 21, 26), social self-concept (items 2, 7, 12, 17, 22, 27), emotional self-concept (items 3, 8, 13, 18, 23, 28), and physical self-concept (items 5, 10, 15, 20, 25, 30). The scores are obtained through a 5-point Likert scale where 1 = never and 5 = always. In this case, for the social dimension, a Cronbach’s alpha score of α = 0.856 was obtained, while for the physical dimension, a value of α = 0.814 was obtained.

Physical Activity Questionnaire for Adolescents [26]: The version adapted to Spanish [27] was used for the present study. This instrument is an instrument that assesses the physical activity carried out by the adolescent during the last 7 days. This questionnaire is made up of a total of 9 questions that are assessed using a 5-point Likert scale. The final score is obtained through the arithmetic mean obtained in the first 8 questions, since the question assesses whether the adolescent was ill or for some reason unable to do physical exercise. In this case, the instrument obtained a degree of reliability of α = 0.803.

State Survey on Drug Use in Secondary Education [28]: This instrument has been promoted by the Spanish Government Delegation for the National Plan on Drugs (PNSD). This survey assesses the use of alcohol (α = 0.771), tobacco (α = 0.731), and cannabis (α = 0.756) by categorizing the frequency of use of these substances into three levels: regular use, occasional use, and no use. In this case, internal consistency of α = 0.749 was obtained.

### 2.3. Procedure

Firstly, a review of the current state of the question was carried out in different databases such as Web of Science, Scopus, and PubMed in order to gain a deeper understanding of the problem addressed in this research. Subsequently, the Department of Didactics of Musical, Plastic and Bodily Expression contacted the different educational centers through a letter informing them of the different objectives of the research. Once the different educational centers gave their approval, the legal guardians of the young people were contacted and informed of the nature and objectives of the present research, asking for their informed consent so that the children could participate in the present study. Once informed consent was obtained from all participants, the questionnaire was administered in the presence of the researchers in order to resolve any doubts that the young people might have. To ensure that participants did not respond randomly, three questions from the different questionnaires were duplicated. As a result, only nine questionnaires were eliminated as they were not filled in properly. Finally, regarding ethical principles, the present study followed the principles established in the Helsinki Declaration of 1975 and was approved and supervised by an ethics committee of the University of Granada (1230/CEIH/2020).

### 2.4. Data Analysis

For the study of the results, the statistical software IBM SPSS Statistics 25.0 program (IBM Corp, Armonk, NY, USA) was used. In this case, a comparative analysis was carried out using the one-factor ANOVA test. The Pearson’s chi-squared test was used to determine statistically significant differences, and the degree of significance was set at 95%. At the same time, Cohen’s standardized d-index [29] was used to calculate statistical power, with effects interpreted as null (≤0.19), small (0.20–0.49), medium (0.50–0.79), and large (≥0.80). The normality of the sample was also studied using the Kolmogorov–Smirnov test, and a normal distribution was found.

The IBM SPSS Amos 26.0 program (IBM Corp, Armonk, NY, USA) was used to develop the structural equation models. This structural model allows for the study of the relationships between the variables that make up each model according to the gender of the participants. In this case, each model is composed of three endogenous variables (PA, P-SC, S-SC) and three exogenous variables (ALC, TBC, CNB) (Figure 1). For the endogenous variables, the causal relationships were examined using the observed associations between the degree of reliability of the measurements and the indicators as references, so that the error caused by the measurement of the observed variables could be included. A significance level of 0.05 was also established. 

Likewise, to obtain an adequate model fit, the goodness-of-fit should be assessed using the chi-squared test, with non-significant *p*-values representing a good model fit. For the comparative fit index (CFI), goodness-of-fit index (GFI) and incremental reliability index (IFI) values above 0.90 indicate a good fit; however, for the root mean square approximation (RMSEA), scores below 0.10 indicate a good fit [30,31].

## 3. Results

To answer H.2. and H.3., Table 1, Table 2 and Table 3 were developed. In this case, Table 1 shows the analysis carried out for alcohol consumption. It can be seen that people who regularly consume alcoholic drinks (M = 4.38) show higher scores than those who consume alcohol occasionally (M = 4.01) or not at all (M = 3.97). Likewise, for physical self-concept, better scores were observed for those who regularly consume alcoholic beverages (M = 4.28), compared to those who consume alcohol occasionally (M = 3.21) or not at all (M = 3.41). For physical exercise, better scores were observed for participants who do not drink alcohol (M = 1.55) than for those who have occasional (M = 1.52) or usual (M = 1.33) consumption.

Table 2 shows the comparative analysis for tobacco use. It was observed that, for the social self-concept, usual use (M = 4.18) reported higher scores than occasional (M = 3.94) or no use (M = 3.96). Likewise, for the physical self-concept, it was observed that non-consumption of tobacco (M = 3.38) reflected better scores than usual (M = 3.25) or occasional (M = 3.19) consumption. For physical activity, it was observed that adolescents who usually smoke (M = 1.59) had higher scores than those who report occasional smoking (M = 1.57) or those who do not smoke (M = 1.51).

Table 3 shows the comparative analysis for cannabis use. In this case, it was observed that, for the social self-concept, the usual use of cannabis (M = 4.33) reflected better scores than occasional use (M = 2.83) or no use of cannabis (M = 3.99). For physical self-concept, it was shown that participants who report not using the substance (M = 3.37) scored higher than those who report occasional (M = 2.41) or usual (M = 3.33) use. For the practice of physical exercise, better scores were observed for those participants who claim to use usually (M = 2.05) compared to those who use occasionally (M = 1.50) or not at all (M = 1.51).

To answer H.1., H.4., and H.5., structural equation models were proposed. The model developed for the entire sample showed a good fit for each of its component indices. For the model developed, the chi-squared analysis showed a significant value (X^2^ = 1.924; df = 4; pl = 0.147). These data cannot be interpreted in isolation due to the sensitivity of the sample size of this research [32]. According to the authors [32], the following indices were used: Comparative Fit Index (CFI), Normalized Fit Index (NFI), and Incremental Fit Index (IFI), each obtaining a score of 0.996, 0.989, and 0.990, respectively. The Tucker–Lewis Index (TLI) was also calculated, showing a score of 0.983, along with the root mean of square error of approximation (RMSEA) showing a value of 0.015.

Figure 2 and Table 4 show the relationships for the whole sample. In this case, for alcohol consumption (ALC), positive relationships were observed with social self-concept (S-SC) (r = 0.049), with tobacco (*p* ≤ 0.001; r = 0.538) and cannabis (CNB) consumption (*p* ≤ 0.001; r = 0.369); however, a negative relationship was shown with physical activity (PA) (*p* ≤ 0.05; r = −0.136). For the tobacco variable (TBC), a positive relationship was observed with social self-concept (S-SC) (r = 0.048), physical activity (PA) (*p* ≤ 0.001; r = 0.059), and cannabis use (CNB) (*p* ≤ 0.001; r = 0.373); however, a negative association was observed with physical self-concept (P-SC) (r = −0.019). Likewise, cannabis use (CNB) showed a positive relationship with physical activity (*p* ≤ 0.05; r = 0.074) and a negative association with physical self-concept (r = −0.068). For physical activity practice (PA), a negative relationship was observed with social self-concept (S-SC) (r = −0.014) and physical self-concept (*p* ≤ 0.001; r = −0.329).

Focusing on the model developed for the male gender (*n* = 396), it showed a good fit for the different indices that make up the model. The chi-squared analysis showed a non-significant *p*-value (X^2^ = 10.625; df = 3; pl = 0.059). The Comparative Fit Index (CFI), Normalized Fit Index (NFI), Incremental Fit Index (IFI), and Tucker–Lewis Index (TLI) obtained values of 0.925, 0.903, 0.909, and 0.900, respectively. Likewise, the root mean of square error of approximation (RMSEA) obtained a value of 0.043. 

Figure 3 and Table 5 show the relationships for the male sample. For alcohol consumption (ALC), a positive relationship was observed with social self-concept (S-SC) (r = 0.023) along with tobacco (TBC) (*p* ≤ 0.001; r = 0.507) and cannabis (CNB) (*p* ≤ 0.001; r = 0.347); however, a negative relationship was observed with physical activity (PA) (r = −0.089). Continuing with tobacco use (TBC), positive relationships were obtained with social self-concept (S-SC) (r = 0.063), physical self-concept (P-SC) (r = 0.067), physical activity (PA) (r = 0.018), and cannabis use (CNB) (*p* ≤ 0.001; r = 0.383). For cannabis use (CNB), there was a negative relationship with physical self-concept (P-SC) (*p* ≤ 0.05; r = −0.120) and a positive link with physical activity (PA) (*p* ≤ 0.05; r = 0.137). Finally, for physical activity practice (PA), negative relationships with social self-concept (S-SC) (*p* ≤ 0.05; r = −0.105) and physical self-concept (P-SC) (*p* ≤ 0.05; r = −0.155) were observed.

Continuing with the model developed for the female gender (*n* = 310), it showed a good fit for the different indices that make up the model. The chi-squared analysis showed a non-significant *p*-value (X^2^ = 11.625; df = 3; pl = 0.024). The Comparative Fit Index (CFI), Normalized Fit Index (NFI), Incremental Fit Index (IFI), and Tucker–Lewis Index (TLI) obtained values of 0.911, 0.904, 0.915, and 0.895, respectively. Likewise, the root mean of square error of approximation (RMSEA) obtained a value of 0.039.

Figure 4 and Table 6 show the relationships obtained for the female gender. In this case, alcohol consumption (ALC) showed positive relationships with social self-concept (S-SC) (r = 0.074), cannabis consumption (*p* ≤ 0.001; r = 0.409), and tobacco consumption (TBC) (*p* ≤ 0.001; r = 0.573); however, a negative relationship was shown with the practice of physical activity (*p* ≤ 0.05; r = −0.198). Continuing with tobacco use (TBC), we found a positive relationship with social self-concept (S-SC) (r = 0.047) along with cannabis use (CNB) (*p* ≤ 0.001; r = 0.381) and physical activity (PA) (r = 0.008); however, negative relationships were shown with physical self-concept (P-SC) (r = −0.041). Continuing with cannabis use (CNB), a positive relationship was shown with physical activity (PA) (r = 0.092) and a negative relationship with physical self-concept (P-SC) (r = −0.063). Finally, for physical activity practice (PA), a negative relationship with physical self-concept (P-SC) (*p* ≤ 0.001; r = −0.372) and a positive link with social self-concept (S-SC) (r = 0.064) was obtained.

## 4. Discussion

The present research showed the relationships between the consumption of harmful substances and their effect on the social and physical dimensions of self-concept and the practice of physical activity in a sample of adolescent students in compulsory secondary education. The obtained results respond to the objectives and hypotheses initially proposed; therefore, the aim of the present discussion is to compare the results obtained with those of other research carried out previously.

The comparative analysis shows that adolescents who regularly use alcohol, tobacco, and cannabis had higher scores on the social dimension of self-concept. Given these findings, it is claimed that these substances are used by adolescents as an element that reaffirms their popularity within the group, reinforcing their position within the peer group [33]. It has also been found that the non-consumption of these substances is associated with lower levels of the social dimension [34].

Very different results were found for the physical dimension. In this case, participants who usually consume alcoholic beverages showed higher scores than those who consume alcoholic beverages occasionally or not at all, while for tobacco and cannabis, those who do not consume alcohol obtained higher scores. Given these results, numerous studies have found that alcohol consumption is widespread in a positive way in different societies and is used as a reinforcing element for physical exercise [35,36]. With regard to tobacco and cannabis, it has been shown that these substances play a negative role on the mental and physical image, with negative repercussions on physical self-concept [37].

Continuing with the practice of physical activity, higher levels of physical exercise were observed for those participants who showed a usual consumption of cannabis and tobacco, while for alcohol, higher scores were obtained from participants who do not consume this substance. Williams et al. [38] and Courtney et al. [39] state that young adolescents are aware of the many benefits of an active lifestyle on physical and mental health, but they are not aware of the consequences of an unhealthy lifestyle on their health. Furthermore, education plays a key role in preventing such attitudes, where a comprehensive educational approach can help prevent such behaviors [40,41,42].

Following the proposed equation models, we observed that alcohol consumption has a positive impact on the social dimension and the consumption of other harmful substances for both sexes. Because of these results, it has been shown that alcohol is widespread as an approved element in different societies and is used as an instrument to reaffirm the social position of individuals [43]. However, these results contrast with those obtained by the Spanish Ministry of Health [28], which states that since 2018, there has been an increase in acute alcohol intoxication in all age groups and both sexes. Likewise, it has also been found that alcohol consumption can lead to the initiation of the consumption of other substances harmful to health such as tobacco and cannabis, with alcohol acting as a catalyst in the consumption of these substances [44]. In contrast, a negative relationship has been found between tobacco consumption and physical activity, with physical exercise having a negative impact on unhealthy behavior [45,46]. 

Continuing with tobacco use, a positive relationship with physical activity and social self-concept was observed. Very distant results were found by Gebert et al. [47], stating that tobacco and cannabis use impairs the attitude towards physical activity as it harms the fitness and health of individuals. However, for the social self-concept, it is stated that tobacco together with alcohol reaffirms people’s social role [48], with such behavior being especially predominant in the male sex [43]. In contrast, a positive relationship between tobacco consumption and physical self-concept was observed for males, while this link was negative for females. In view of these findings, very distant results were obtained by García-Canto et al. [49], who affirmed that tobacco consumption harms people’s fitness and therefore their physical self-image. Negative relationships have also been found between cannabis and physical self-concept, as this psychoactive substance has a detrimental effect on health, acting on numerous areas of the brain [50,51].

Finally, the relationship between the social and physical dimensions of self-concept showed a negative relationship with the practice of physical activity for the male sex; however, for the female sex, a positive relationship was observed between social self-concept and the practice of physical exercise. Very different results were found by Ramírez-Granizo et al. [52], who stated that during adolescence, the female sex becomes detached from the practice of physical activity, preferring to carry out other tasks of a more sedentary nature. In perspective with the results found, Sanz-Martín et al. [53] and Peláez-Barrios and Vernetta-Santana [54] state that regular physical exercise has academic, emotional, social, family, and physical benefits. 

## 5. Limitations and Future Perspectives

Although this research shows the existing relationships between the study variables, it has a series of limitations which should be highlighted. 

In this case, the design of the research reflects one of the limitations since, as it is cross-sectional in nature, it only allows us to establish the relationships between the variables at that moment in time and not over a longitudinal period. Likewise, the participants lived in the south of Spain, so it is not possible to establish generalizations in a wider area of the national geography. Another limitation to be pointed out is the use of the instruments employed, which, despite being validated by the scientific community, have an intrinsic measurement error. 

Continuing with future perspectives, the development of a project to train future physical education teachers from a healthy point of view is being carried out.

## 6. Conclusions

The comparative analysis disproves hypothesis number two, as it was observed that non-consumption of cannabis and tobacco reported higher scores in the physical self-concept; however, regular consumption of alcoholic beverages reported higher scores in this area of the self-concept. On the contrary, hypothesis three was confirmed since it was observed that regular consumption of alcoholic beverages together with regular consumption of tobacco and cannabis reported higher scores for the social area of self-concept. 

With respect to the structural equation models, hypothesis number one was confirmed, as differences were found between the male and female sexes. In this case, a negative relationship was observed between the practice of physical activity and physical self-concept in the three models proposed, not complying with hypothesis number four. In addition, it was observed that the total population and the male population showed a negative relationship between social self-concept and the practice of physical activity, with a positive relationship for the female sex, with these results being distant from those expected in hypothesis number five. It was also observed that the relationship between social self-concept and alcohol and tobacco consumption was positive for both sexes, and the same was true for physical self-concept and cannabis consumption; however, for physical self-concept and tobacco consumption, a positive relationship was shown for males and a negative one for females. 

Finally, this research highlights the importance of the educational sphere, since young people should be educated from an active and healthy perspective. In order to carry out this type of education, the area of physical education plays a fundamental role, as it is through physical education that a positive motivation towards active and healthy lifestyles can be developed.

## Figures and Tables

**Figure 1 ijerph-19-10243-f001:**
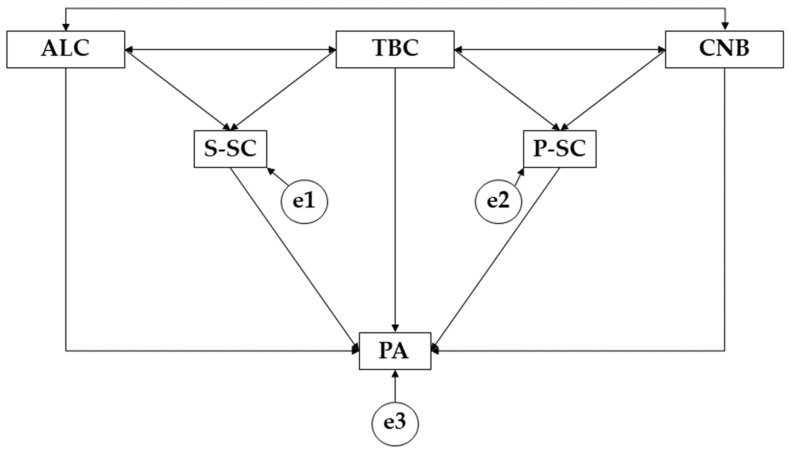
Structural equation model. Note: alcohol (ALC); tobacco (TBC); cannabis (CNB); social self-concept (S-SC); physical self-concept (P-SC); physical activity (PA).

**Figure 2 ijerph-19-10243-f002:**
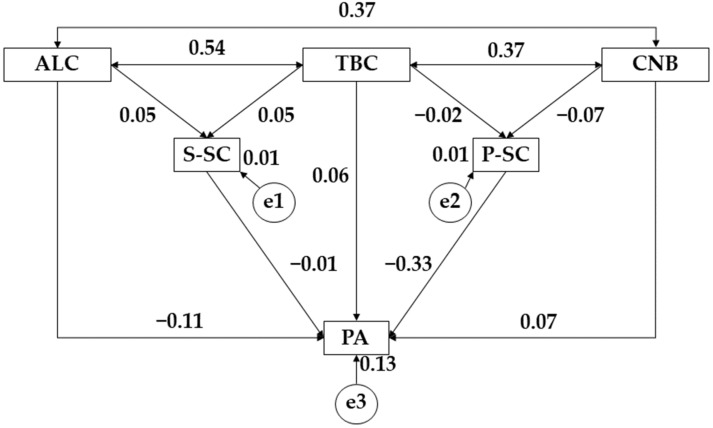
Proposed structural equation for the general survey population. Note: alcohol (ALC); tobacco (TBC); cannabis (CNB); social self-concept (S-SC); physical self-concept (P-SC); physical activity (PA).

**Figure 3 ijerph-19-10243-f003:**
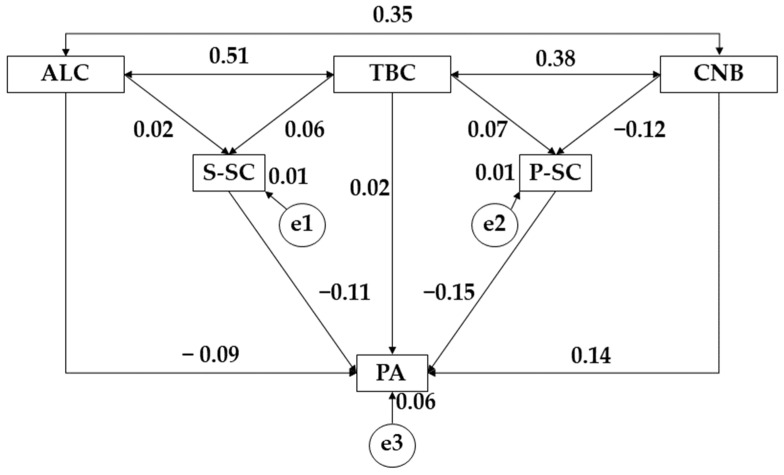
Proposed structural equation for the male population. Note: Alcohol (ALC); tobacco (TBC); cannabis (CNB); social self-concept (S-SC); physical self-concept (P-SC); physical activity (PA).

**Figure 4 ijerph-19-10243-f004:**
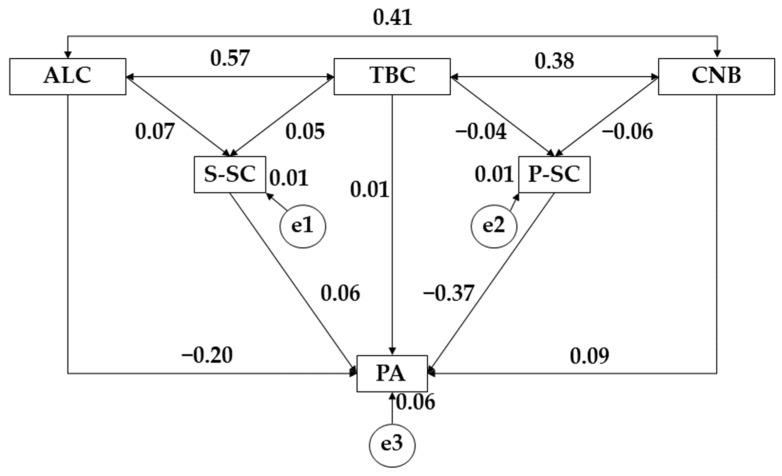
Proposed structural equation for the female population. Note: Alcohol (ALC); tobacco (TBC); cannabis (CNB); social self-concept (S-SC); physical self-concept (P-SC); physical activity (PA).

**Table 1 ijerph-19-10243-t001:** Comparative analysis of alcohol consumption.

		N	M	SD	F	*p*	ES (d)	95% CI
S-SC	Usual consumption	6	4.38	0.479	1.637	>0.05	NP	NP
	Occasional consumption	310	4.01	0.748				
	No consumption	390	3.97	0.688				
P-SC	Usual consumption	6	4.28	0.479	2.912	≤0.05	0.768 ^a^ 0.798 ^b^	[0.228; 0.847] ^a^ [0.368; 0.994] ^b^
	Occasional consumption	310	3.21	0.911				
	No consumption	390	3.41	0.842				
PA	Usual consumption	6	1.33	0.516	1.933	>0.05	NP	NP
	Occasional consumption	310	1.52	0.504				
	No consumption	390	1.55	0.497				

Note 1: Social self-concept (S-SC); physical self-concept (P-SC); physical activity (PA). Note 2: ^a^ Differences between usual consumption and no consumption; ^b^ differences between usual consumption and occasional consumption.

**Table 2 ijerph-19-10243-t002:** Comparative analysis of tobacco consumption.

		N	M	SD	F	*p*	ES (d)	95% CI
S-SC	Usual consumption	44	4.18	0.636	1.691	>0.05	NP	NP
	Occasional consumption	198	3.94	0.783				
	No consumption	464	3.96	0.682				
P-SC	Usual consumption	44	3.25	0.913	1.710	>0.05	NP	NP
	Occasional consumption	198	3.19	0.879				
	No consumption	464	3.38	0.829				
PA	Usual consumption	44	1.59	0.497	0.827	>0.05	NP	NP
	Occasional consumption	198	1.57	0.500				
	No consumption	464	1.51	0.501				

Note 1: Social self-concept (S-SC); physical self-concept (P-SC); physical activity (PA). Note 2: Differences between usual consumption and no consumption; differences between usual consumption and occasional consumption; differences between occasional consumption and no consumption.

**Table 3 ijerph-19-10243-t003:** Comparative analysis of cannabis consumption.

		N	M	SD	F	*p*	ES (d)	95% CI
S-SC	Usual consumption	8	4.33	0.454	3.640	≤0.05	0.324 ^a^	[0.015; 0.634] ^a^
	Occasional consumption	40	2.83	0.769				
	No consumption	658	3.99	0.694				
P-SC	Usual consumption	8	3.33	0.512	1.664	>0.05	NP	NP
	Occasional consumption	40	2.41	0.9776				
	No consumption	658	3.37	0.869				
PA	Usual consumption	8	2.05	0.156	1.512	>0.05	NP	NP
	Occasional consumption	40	1.50	0.502				
	No consumption	658	1.51	0.783				

Note 1: Social self-concept (S-SC); physical self-concept (P-SC); physical activity (PA). Note 2: ^a^ Differences between usual consumption and no consumption; differences between usual consumption and occasional consumption; differences between occasional consumption and no consumption.

**Table 4 ijerph-19-10243-t004:** Structural model for the general study population.

Associations between Variables	R.W				S.R.W
	Estimates	S.E	C.R	*p*	Estimates
S-SC ← ALC	0.024	0.022	1.106	0.269	0.049
S-SC ← TBC	0.021	0.019	1.086	0.278	0.048
P-SC ← TBC	−0.010	0.022	−0.469	0.639	−0.019
P-SC ← CNB	−0.082	0.049	−1.680	0.093	−0.068
PA ← TBC	0.019	0.013	1.380	0.167	0.059
PA ← S-SC	−0.010	0.026	−0.404	0.686	−0.014
PA ← P-SC	−0.189	0.020	−9.328	***	−0.329
PA ← CNB	0.051	0.027	1.898	**	0.074
PA ← ALC	−0.049	0.015	−3.182	**	−0.136
TBC ← → ALC	0.538	0.095	12.584	***	0.538
TBC ← → CNB	0.373	0.046	9.285	***	0.373
ALC ← → CNB	0.369	0.040	9.195	***	0.369

Note 1: Alcohol (ALC); tobacco (TBC); cannabis (CNB); social self-concept (S-SC); physical self-concept (P-SC); physical activity (PA). Note 2: Regression weights (R.W); standardized regression weights (S.R.W); estimation error (S.E); critical ratio (C.R). Note 3: ** *p* < 0.05; *** *p* < 0.001.

**Table 5 ijerph-19-10243-t005:** Structural model for male population.

Associations between Variables	R.W				S.R.W
	Estimates	S.E	C.R	*p*	Estimates
S-SC ← ALC	0.011	0.028	0.395	0.693	0.023
S-SC ← TBC	0.027	0.025	1.078	0.281	0.063
P-SC ← TBC	0.038	0.031	1.245	0.213	0.067
P-SC ← CNB	−0.133	0.060	−2.220	**	−0.120
PA ← ALC	−0.031	0.020	−1.535	0.125	−0.089
PA ← CNB	0.085	0.033	2.529	**	0.137
PA ← S-SC	−0.078	0.036	−2.152	**	−0.105
PA ← P-SC	−0.086	0.027	−3.152	**	−0.155
PA ← TBC	0.006	0.019	0.311	0.756	0.018
TBC ← → ALC	1.043	0.116	8.988	***	0.507
TBC ← → CNB	0.448	0.063	7.115	***	0.383
ALC ← → CNB	0.364	0.056	6.519	***	0.347

Note 1: Alcohol (ALC); tobacco (TBC); cannabis (CNB); social self-concept (S-SC); physical self-concept (P-SC); physical activity (PA). Note 2: Regression weights (R.W); standardized regression weights (S.R.W); estimation error (S.E); critical ratio (C.R). Note 3: ** *p* < 0.05; *** *p* < 0.001.

**Table 6 ijerph-19-10243-t006:** Structural model for the female population.

Associations between Variables	R.W				S.R.W
	Estimates	S.E	C.R	*p*	Estimates
S-SC ← ALC	0.038	0.036	1.076	0.282	0.074
S-SC ← TBC	0.021	0.030	0.686	0.493	0.047
P-SC ← TBC	−0.020	0.030	−0.674	0.501	−0.041
P-SC ← CNB	−0.078	0.076	−1.025	0.305	−0.063
PA ← S-SC	0.039	0.032	1.225	0.221	0.064
PA ← CNB	0.063	0.040	1.580	0.114	0.092
PA ← ALC	−0.062	0.021	−3.019	**	−0.198
PA ← TBC	0.002	0.017	0.117	0.907	0.008
PA ← P-SC	−0.208	0.029	−7.155	***	−0.372
ALC ← → CNB	0.383	0.057	6.657	***	0.409
TBC ← → ALC	1.372	0.157	8.741	***	0.573
TBC ← → CNB	0.381	0.067	6.254	***	0.381

Note 1: Alcohol (ALC); tobacco (TBC); cannabis (CNB); social self-concept (S-SC); physical self-concept (P-SC); physical activity (PA). Note 2: Regression weights (R.W); standardized regression weights (S.R.W); estimation error (S.E); critical ratio (C.R). Note: ** *p* < 0.05; *** *p* < 0.001.

## Data Availability

The data used to support the findings of current study are available from the corresponding author upon request.

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
