# Peer review of "Alcohol, Tobacco and Cannabis Consumption on Physical Activity and Physical and Social Self-Concept in Secondary School Students: An Explanatory Model Regarding Gender"

_ijerph, 2022, doi:10.3390/ijerph191610243_

Round 1

Reviewer 1 Report

-          Because usually keywords don't take (over) sequences from the title, (is possible) can you please replace them - so can reflect the ideas in the article and not just be redundant?

-          The introduction should include in more detail the gap in existing literature and the innovative aspects brought by this paper - analysis for existing literature and the novelty brought by this paper should be highlighted

-        The section of introduction should include (even briefly at the end of the chapter): the context of the study, which are the main results presented  in short, which is the originality of this paper, the main implication policy of these results and a description of the structure of the paper, detailing the role of each section of the paper. Some of them are missing - please fill it accordingly

-        The hypotheses should be more specific and reflect a statement to be validated or invalidated by the research

-          The interpretation of the data in the Tables and Figures should be further mentioned in the paper. In addition, correlation with the three hypotheses should be highlighted (which data refers to which hypothesis) mentioning also if they are validated or invalidated.

-          An area related to the clear / practical / effective applicability of the study and an area of concrete proposals (some exist in the body of the text) should be highlighted in Conclusion chapter

Author Response

Dear Reviewer, 

I am attaching our answers.

Thank you!

Reviewer 2 Report

This is an inspired empirical study of a social problem affecting adolescents and social substance abuse and the effects of these substances on the vocational and cognitive abilities of the person.  The research background, hypotheses, research design, data collection and analysis (SPSS Statistics 25.0 and Amos 26.0) is appropriate to the research project and informs the conclusions of the study.  The references cited are appropriate and current and lend validity to the research.  This research adds to the literature substantially. 

One issue on Line 129: SPSS Statics, or SPSS Statistics?

Author Response

(The authors gave the same response as above.)
